# Barriers and facilitators in physical activity among youth with cerebral palsy in Sweden: A qualitative study

Frida Degerstedt[1,2]*, Britt-Inger Keisu[3], Martin Björklund[1,4], Birgit Enberg[1]

1 Department of Community Medicine and Rehabilitation, Physiotherapy, Umeå University, Umeå, Sweden, 2 Umeå Center for Gender Studies, Umeå University, Umeå, Sweden, 3 Department of Social and Psychological Studies, Karlstad University, Karlstad, Sweden, 4 Centre for Musculoskeletal Research, Department of Occupational Health Sciences and Psychology, University of Gävle, Gävle, Sweden

* Frida.degerstedt@umu.se

## Abstract

Limitations to full social participation in physical activity—both during leisure time and in physical education—persist among young individuals with cerebral palsy. This study explores the experiences of physical education and physical leisure activities among youth with cerebral palsy living in Sweden. It examines how these youths navigate contextual barriers and facilitators to participation, focusing on norms of function and gender. Individual interviews with 13 youth, aged 15–18 years, were analysed using Qualitative Content Analysis resulting in a main theme: Being in a continuum between exclusion and empowerment, with three categories: 1) Excluded and denied of support; 2) Resisting prejudice and discrimination; and 3) Empowered, with appropriate support. Most participants had experienced not being given equal opportunity in particularly in physical education but also in leisure activities and had encountered limiting norms and preconceptions based on gender and function. The findings indicate that norm-consciousness, respect and involvement in decision-making are key for participation and empowerment.

## Introduction

In this study, we delve into the stories of youth with cerebral palsy and relate this to the fundamental right to engage in physical activities in general, including physical education in school. Cerebral palsy is the most common cause of motor impairments among children and young persons in the Western world [1–3], often accompanied by functional differences in cognition, perception, or communication [4].

Youth and adults with cerebral palsy report significant physical, psychological, and social benefits from engaging in physical activity [5–7]. Nevertheless, in accordance with youth in general, they are far from reaching the World Health Organization's [8] recommendations for physical activity [9]. Motor function per se can affect a young person's ability to participate in physical activities, including factors such as pain

**Data availability statement:** The dataset generated and analyzed during this study contains potentially identifying or sensitive patient information. Due to ethical restrictions, and to protect participant confidentiality, the minimal data can not be shared publicly. Data for this study are available upon request from Umeå University, Department of Community Medicine and Rehabilitation, via email (studieadm.sam-rehab@umu.se) for researchers who meet the criteria for access to confidential data.

**Funding:** This work was supported by: MB: the Linnea and Josef Carlsson Foundation; MB the Strategic Research Area Health Care Science (SFO-V), Umeå University; FD: The Queen Silvia's Jubilee foundation 2019; FD: Umeå Center for Gender Studies at Umeå University. The funders had no role in study design, data collection and analysis, decision to publish, or preparation of the manuscript.

**Competing interests:** The authors have declared that no competing interests exist.

or fatigue. Our previous study showed that youth with greater gross motor difficulties tend to be less involved in both physical education and leisure activities [10]. However, it is important to recognize that low expectations and a lack of accessible, inclusive opportunities also play a major role [11]. Discrimination can lead to significant social barriers that limit participation. Therefore, understanding young people's conditions for engaging in physical activity in general, such as in leisure time and in physical education in school, requires not only an individual perspective, but also a societal one that considers broader social and structural factors.

In this paper, we use the term 'function' to refer to the overall meaning attributed to bodily or cognitive traits and differences, whereas 'disability' is used to describe limiting contextual processes or to reflect the terminology employed by others. According to Garland-Thomson [12], disability is usually defined as functional departures from the norm. Meanwhile, the narrow descriptions of the perceived non-disabled person are also constructed as a social profile that is practically unattainable [12]. This perspective helps illuminate how participation in physical education and physical activity is shaped not only by physical ability, but also by broader societal structures—including accessibility, institutional support, and socially constructed expectations. These normative frameworks can offer a sense of identity and belonging, yet they may simultaneously restrict individual freedom of choice and contribute to inequity [13]. One example of this is that the male, able-bodied, sports-active student tends to be considered the norm in sports and physical education, reflecting both ableist and gendered assumptions about who is expected to participate and excel in these contexts [14]. This may limit participation, creating barriers for those whose bodies or identities fall outside dominant ideals.

To further understand these barriers, previous research has explored how perceptions of disability influence participation. Previous research has also shown that the perception of impaired bodies as flawed and lacking abilities becomes a barrier to engaging in physical activity [15], a phenomenon also presented in the field of physical education [16,17]. Meanwhile, physical ability in itself is not typically perceived as a major barrier by youth with cerebral palsy, as long as they are not compared to presumed able-bodied youth [18,19]. Emphasizing individual abilities, rather than deficits, can therefore foster more inclusive practices. Such knowledge may be explored through youths' own experiences of physical activity participation.

Despite these challenges, the Swedish school system is built on principles of inclusion and equal access to education for all. In Sweden, all children and youth are entitled to adapted, inclusive education regardless of school form or motor difficulties. School attendance is mandatory in Sweden until the 9th grade, and cost free throughout upper secondary school [20]. The school subject of physical education is no exemption. Physical education aims to develop students' practical and theoretical knowledge of various physical activities, outdoor life, and healthy lifestyles [20]. Despite the legal right to inclusion, youth with cerebral palsy who have more gross motor difficulties participate less also in physical education [10], which may indicate insufficient adaptations within teaching and the school setting. Research highlights several interacting factors that influence participation in physical education

or physical leisure activity, including personal factors such as pain, skills, and confidence/self-efficacy [21–24], and environmental factors such as technical aids or accessibility, and adult/peer support or inclusivity [11,19,21,22,24–26]. Additionally, various barriers are sustained by attitudinal and financial constraints [27]. Meanwhile, what children and youth describe as important for physical activity participation, is the social interaction and having fun [5,28].

Considering the joy and health benefits associated with physical activity [8,29], and the inequality incorporated as the norm in traditional sports, which benefits some bodies and marginalizes others [30], it is of interest to illustrate youths' own experiences of physical leisure activity and physical education participation. Against this backdrop, this study adds to this knowledge by aiming to explore the experiences of physical education and physical leisure activities among youth with cerebral palsy. Specifically, it investigates how these youths navigate contextual barriers and facilitators to participation, with particular attention to norms related to function and gender.

With the current study, we seek to enhance theoretical and empirical knowledge of youths' own perspectives which are insufficiently shared. We provide a norm critical perspective to explain how youth describe the expectations associated with growing up and living with a cerebral palsy diagnosis regarding physical education and physical leisure activity. This knowledge is essential for understanding the underlying factors contributing to the low levels of physical activity among individuals with cerebral palsy compared to WHO recommendations. In a broader context, it may contribute to insight into determinants of societal participation, experiences of discrimination, and health. By focusing on what is considered the norm ('normal'), by youth and those around them, we can illuminate what is not, and how such conceptions influence the lives of young people with cerebral palsy. Knowledge about who gains privileges from the prevailing norms—and who does not—may have significance for other groups in society that fall outside the general norms and expectations for various reasons.

## Conceptual framework

Building on Garland-Thomson's [31] theory, this article applies four modes of representation, based on how disability is perceived in American public photography, to create a conceptual framework for pinpointing contextual barriers and facilitators to participation among youth with cerebral palsy. As part of this framework, we also consider how gender and disability intersect in shaping experiences of barriers and facilitators to participation. Although norms surrounding disability and gender are often constructed in intertwined ways—shaped by overlapping social and cultural processes—they can still be analytically distinguished. In our study, both gender and (dis)ability are understood as socially and culturally (re)constructed concepts. Gender is not understood as a fixed attribute or inherent identity, but rather as an ongoing social process—something one *does*, not something one *is*. Gender is enacted through everyday interactions and shaped by social norms, expectations, and institutional contexts. It is relational to biological sex but varies across time and space, making it a dynamic and context-dependent construct [32]. Similarly, notions of ability—and of the "non-disabled"—are continuously renegotiated and shaped by context [32].

Garland-Thomson [12] describes disability as a construct within a system of exclusion that stigmatizes human differences. She identifies how disability is commonly depicted, which we found useful also in the context of physical activity for youth who have cerebral palsy in Sweden: The four modes are *the exotic, the wondrous, the sentimental*, and *the realistic modes* [31]. They reflect a history of disabled people being on display, mirroring the contexts in which they live. The comparison of bodies legitimizes the unequal distribution of status, resources, and power in the social and built environment.

The first mode, *the exotic*, makes disability distant and strange, emphasizing sensationalism and objectification without involving the viewer. It does not invoke astonishment or pity, but rather alienation and contrast from the viewer. The second mode is *the wondrous*, which highlights supposedly amazing achievements to create admiration and astonishment. Even simple activities, like eating, may be seen through the wondrous mode, portraying the disabled person as courageous and overcoming disability. This mode emphasizes the distinction between the viewer and the person considered disabled. The third mode is *the sentimental*, which places the disabled person in a diminished position, as helpless,

suffering, and evoking pity. This mode infantilizes the disabled person, constructing someone who needs care, thereby strengthening the viewer's sense of progress and power. Finally, the fourth mode is *the realistic*, which minimizes the difference between the viewer and the disabled person. In this mode, the viewer is supposed to identify with the disabled person, regardless of the impairment. Realism does not depict reality but rather an illusion of it, constructed like the other modes [31].

## Method

A qualitative design based on semi-structured individual interviews was used for the current study [33]. Criteria for reporting qualitative studies (COREQ) was followed [34]. The current study has followed the principles of the Declaration of Helsinki [35]. Ethical approval has been obtained from the Regional Ethical Review Board in Umeå (2018/219–31) and the Swedish Ethical Review Authority (2020/01280). All participants were informed about the study, verbally and in writing. Informed consent was obtained from all the participants. Consent from parents or guardians is not required from the age of 15, according to the Swedish Ethical Review Authority, provided that the participants are able to understand their engagement in research and provide informed consent. All participants were deemed capable to provide informed consent. Participants were invited to share information about the study with their guardians, though this remained at their discretion.

## Participants

The data originates from thirteen participants, all born and raised in Sweden: five identified as girls and eight as boys. Participants' physical abilities varied widely—from very limited movement, requiring help to steer an electric wheelchair, to only minor difficulties, such as a slight limp or slower running speed. Most participants had experiences from both integrated physical education classes and classes adapted specifically for pupils with physical impairments. The participants all described present or previous participation in physiotherapy, which, while not specifically included in the current paper, may be intertwined with physical leisure activities, and sometimes even with physical education. All participants engaged to some degree in physical leisure activity, though few were currently participating in organized sports. All participants were in school and had participated in physical education, but some did not have physical education in their educational program schedule at the time of the interviews. Inclusion criteria were: being 15–18 years of age, having a diagnosis of cerebral palsy and being capable of providing informed consent to participate in research. Moreover, while not an explicit criterion, the use of language or communication techniques not familiar to the interviewer or requiring training, was discussed with the recruiting care providers as a potential risk of misinterpretation of the participants. None of the volunteering youths that were referred to us were excluded. To recruit participants, the first author contacted child rehabilitation units in 20 out of the 21 Swedish health care regions. Physiotherapists, or other health care providers, provided names and phone numbers of interested youth. Next, the participants were contacted with a text message and/or phone call, offering more information about the study and scheduling of the interview. The participants received verbal and written information, in plain language, about the study and their right to decline or terminate participation at any time. They all consented to participation in the research study, either by signing the consent form or by consenting verbally in the recording before the start of the interviews. The confidentiality of data and the participant's rights were repeatedly emphasized during correspondence and interviews.

## Data collection

Data collection was performed between June 20, 2021, to February 16, 2024. The interviews were performed individually, either digitally through Zoom (n = 6) or face-to-face (n = 7) in a silent place chosen by the participants in discussion with the first author (the interviewer), for example the child rehabilitation unit or participant's home. One participant chose to include a school assistant in the interview situation, for practical and communication support. The semi-structured

 

interviews were performed, recorded, and transcribed by the first author, and lasted for an average of 46 (22–94) minutes. Two participants showed signs of tiredness toward the end of the interviews but chose to continue until the interviews were completed, when asked. The interviews included questions regarding the participants' experiences of participation (or non-participation) in physical leisure activity and physical education in school. These terms were not specifically defined but rather the participants were allowed to describe their views and experiences. Examples of the interview questions were: 'Do you have any examples of when physical education/leisure activity worked out well/not as well?' or 'were there anyone (else) in your class/team who seemed to struggle to participate?'. Questions and answers were followed by additional questions when appropriate, such as 'How did you feel in that situation?', 'why do you think they did that?' Open- as well as closed ended questions were used to facilitate communication as open questions at times may be perceived challenging [36]. The participants were allowed to elaborate outside the interview guide. Moreover, participants were asked for explicit experiences of norms or expectations on how to be and how to perform, where gender, ethnicity/skin colour, and function were understood to be the basis of such expectations. However, only norms related to gender and function were visible in the data.

The first author has several years of experience as a physiotherapist within child- and youth rehabilitation, and of interviewing children, youth and adults as patients, as well as some experience with research interviews. This experience was thought to facilitate the interviews. The current interview guide was tested before the study, with members of a reference group consisting of young adults who have cerebral palsy, physiotherapists within child rehabilitation, and the authors, both during construction and in a staged interview-session. The participants of the current study were aware of the interviewer's background.

## Data analysis

Qualitative content analysis, according to Graneheim and Lundman [33], was used, aided by the data analysis program Max-QDA [37]. This methodology provides a systematic as well as flexible framework to analyse data, allowing latent interpretation meanwhile staying close to data (the participants' narratives). Transcribed interviews were read repeatedly. Next, the text was divided into meaning units, which were condensed and given a code, keeping the context in consideration. Codes were compared and grouped together for groups to form categories and ultimately interpreted into a theme [33] (Table 1).

Trustworthiness of analysis was sought through repeated discussions between first and second authors about codes and code groupings. Discrepancies in the coding procedure that occurred during analysis and between the authors were

**Table 1. Example of the analysis process.**

| Meaning unit | Code | Sub-category | Category | Theme |
|---|---|---|---|---|
| I got to hear that several times, that 'no you have to stand on your feet, or the others may hurt themselves.' And then I'm like 'but what about me? I fall all the time, don't I hurt myself?' | Repeated teacher oppression for the sake of others. | Being perceived as less important | Excluded and denied of support | Being in a continuum between exclusion and empowerment |
| ... like when they see me do things they stare…/… I remember running down the slope in our yard and the neighbors' grandmother was staring at me with wide eyes. | They stare at me when I do physical things | Silence as a barrier | | |
| Anyway, I didn't participate…in this class-day, because I was thinking they won't have patience with me… and it will become so intense… It was such a shame, really, because when I did previously it felt like such a nice community. | Refrains from participating for the sake of the team. | Assuming, responsi-bility and guilt | Resisting prejudice and discrimi-nation | |
| ...we sat down and planned together for... uh, what was possible to carry out, …/… and then he said – here and here we'll do skiing and skating, and I'm thinking this for you, do you think that will work? – yes, I think it will probably work. Just something like that... really helps a lot... | Joint physical education plan-ning improves participation | – | Empowered, with appropri-ate support | |

aligned through re-coding when considered appropriate. How well categories covered data and similarities and differences between categories, were repeatedly discussed between authors, and regrouping of content and renaming of categories were made when considered necessary in order for categories to become pertinent to data. This was followed by triangulation where joint reading of selected interviews by the last two authors to verify the analysis. When there were discrepancies between the co-authors regarding how categories were organized and separated, revisions were made until consensus was reached. Both manifest and latent content were analysed [38].

## Results and discussion

Through the analysis an overarching theme was constructed; Being in a continuum between exclusion and empowerment, which reflects the young people's overall experiences of participation in physical leisure activity and physical education. The theme is comprised of three categories covering interpretations of participants' descriptions of various levels of participation and non-participation: 1) Excluded and denied of support; 2) Resisting prejudice and discrimination; and 3) Empowered, with appropriate support (Table 2).

In this section, we will present our three constructed categories including sub-categories, each followed by a discussion and analysis.

### Category 1. Excluded and denied of support

Within this category, participants described situations with exclusion, where they experienced very limited power to act or resist. The participants' experiences involved exposure to discrimination through different processes of explicit or implicit exclusion, because of how they looked or functioned, and through other people's acts of care, which in certain situations were perceived as diminishing.

**Being perceived as less important.** According to data, comparison of ability was described as 'limiting', as others were seen as superior. Several participants described situations during physical education where they did not seem to matter as much as their peers. Nothing was arranged at times when the class activities did not work out for them, which meant that either the youth themselves had to take responsibility for their education or that they did not receive any.

> Well, sometimes I wasn't allowed to participate [in physical education] …so I just came up with things to do, myself, that I made up at the time. (Participant 9, 18 years)

Coping with such occasions was repeatedly described as resignation due to perceived powerlessness. Being obstructed from physical education participation by not being allowed to use the wheelchair was also described.

**Table 2. Category tree: One overarching theme, and three categories with subcategories, as well as discussion headlines following each category.**

| Being in a continuum between exclusion and empowerment | | |
|---|---|---|
| *1. Excluded and denied of support* | *2. Resisting prejudice and discrimination* | *3. Empowered, with appropriate support* |
| *Being perceived as less important* | *Challenging perceptions of body, and gender* | |
| *Silence as a barrier* | *Assuming responsibility and guilt* | |
| *Derogatory caring* | | |
| ***Exclusionary barriers – Discussion category 1*** | ***Struggle to resist or conform to barriers – Discussion category 2*** | ***Support facilitates participation – Discussion category 3*** |

I got to hear that several times, that "no you have to stand on your feet, or the others may hurt themselves." And then I'm like 'but what about me? I fall all the time, don't I hurt myself?' (Participant 3, 17 years)

A boy had noticed that the teacher put more effort into finding meaningful alternative activities when other pupils were unable to participate in physical education.

…/… but it was like, when I was the only one [who could not participate], then it was like it wasn't as important [to find appropriate alternatives] (Participant 7, 16 years)

Hence, social exclusion due to a lack of engagement and communication from their physical education teachers, was experienced by most participants. This was at times despite repeated attempts from the participants themselves, their parents, or physiotherapists to inform the teachers about the participants' physical condition and suggest adaptations. One participant recalled a cross-country skiing class, where he was assigned a special sled in which he was passively pushed in the ski-trails—something he particularly disliked as he wanted to actively participate. This illustrates that even with adaptations in place, inclusion in decision-making was neglected.

**Silence as a barrier.**  Silence was also part of the process of exclusion that was experienced a barrier to participation. This was, for example, visible when the role of peers was described. While most participants declared that no one had ever explicitly told them not to participate in team activities, one participant added that not being banned is not the same as being invited.

No… no, no-one said it straight out. But nor did they say "hey, are you up for the baseball game tomorrow" (Participant 3, 17 years)

Most participants also described situations where pupils were staring or kept their distance by avoiding eye contact, particularly in new contexts.

When people that don't know me meet me, like, some just look away and think I'm really weird…/…like the first physical education class in high school, I was so sad and upset…/…some may think that I can't do things, and some may think that I don't want to…But I did want to! (Participant 10, 17 years)

One of the interviewed girls described silence towards her from peers and leaders in the football team she had to switch to when her old team was shattered. She assumed this was because of her cerebral palsy but neither peers nor leaders addressed this. Combined with a sense of not doing well enough, this resulted in her quitting the team.

**Derogatory caring.**  Scenarios resulting in low expectations placed on participants due to cerebral palsy were described. Being underestimated was expressed as problematic; for example, through seemingly friendly consideration, or exaggerated praise that was not perceived as proportionate to the accomplishment. This seemed difficult for participants to object to or resist.

Those that are not in this "[disability-] world" …/…can sort of disparage you and say, sort of, well that everything is so good and well done, although you didn't do that much…it is like because they want to make me happy, but it doesn't have to be so exaggerated, it becomes like the other way around (Participant 8, 17 years)

Another participant described receiving a high grade in physical education that she didn't approve of since she had been participated in physical education mostly separated from her class, and the teacher had rarely seen her perform. Hence, she perceived it as a "pity-grade". Another example of such unasked-for consideration was when being excluded from performing physically strenuous things, by peers or relatives who wanted to spare her exertion.

It is like fed into me: "Take it easy! Sit down and rest! Are you in pain today?" instead of "Come on! Get on with it! Yes, it may hurt but you can do it!" A bit like people are tiptoeing around me. (Participant 3, 17 years)

While such consideration may have been well meant, the participant was questioned for being active and deprived of the support and encouragement that she thought she would benefit from.

## Exclusionary barriers—discussion category 1

In the first category, Excluded and denied of support, social barriers leading to explicit or implicit exclusion were observed, such as not being asked to join the class team, being met with silence and disregard, or through seemingly considerate actions. These situations, particularly in physical education, were linked to a lack of teacher engagement and cooperation, aligning with qualitative findings by Shields and Synnot [21]. Such school exclusion can be compared to discrimination. A Swedish study by Ring et al. [39] also describes limited feelings of belonging and opportunity to contribute as barriers to physical education for youth with impairments.

When participants encountered discriminatory behaviours, it created distance and a lack of connection with other teenagers or adults. This aligns with Garland-Thomson [31], who found that disabled individuals are socially constructed as 'different' and viewed through *the exotic mode*, as a bit 'freakish.' Participants also described receiving excessive praise for simple tasks and good grades in physical education despite minimal teacher involvement. Garland-Thomson [31] termed this *the wondrous mode*, where accomplishments are ranked by different standards than those for individuals perceived non-disabled. Participants did not actively resist these patronizing behaviours, possibly because the compliments (merited or not) made them harder to critique, as noted by Wang et al. [40]. Unequal power relations between teenagers and adults, and between participants and non-disabled peers, is another interpretation [41]. While exclusion through caring was seen as a moderate form of exclusion, by the participants, it still limited the participants' agency.

## Category 2: Resisting prejudice and discrimination

The second category includes ways of speaking or reasoning among the participants concerning physical activity that comprised exclusion or discomfort, similar to the first category. However, this category allows room for struggle to resist, challenge, or sometimes defend exclusionary practices, to seek one's place, rather than the powerlessness expressed in the previous category, Expectations from others as well as themselves were negotiated for acceptance.

**Challenging perceptions of body and gender.** This category included the participants' eagerness to find, establish, or advance their social position and identity. Mostly, such positioning was evident in terms of a hierarchical view of function, where the non-impaired body was seen as superior. Comparison was made with those who were presumably further down in such a hierarchy in terms of cognitive or physical functioning.

Well, like this, I do have rather mild [cerebral palsy], it's like my leg and my arm so it's not like… well, I'm not in a wheelchair, which I think is good. (Participant 4, 17 years)

Hierarchical positioning included emphasizing that certain features or functions were not due to their diagnosis, but rather related to expected differences such as a lack of personal talent or, as described below, age-related clumsiness.

Now that I'm 17 and reached my teens, the body grows the most, so, usually it's like this—teenagers are a bit clumsy—but that adds on, onto my problem. (Participant 2, 17 years)

Some participants also positioned themselves according to gender, from the perspective that physical activity traditionally favoured the masculine body. A female participant suggested that gender issues were overridden by function when having cerebral palsy. She argued that she had gotten to try out more different sports activities than other girls because she was

not seen as particularly feminine. The same participant also noted gendered differences within physical education, where boys engaging in ball sports had an advantage in physical education.

> You had an advantage if you were playing hand ball in your leisure [time] because there were *lots* of ball sports [in physical education] …/… well, so it was actually the guys that were doing physical education, and the girls sat on the side or did other things. (Participant 3, 17 years)

Some of the male participants also noticed that girls participated less in physical education than boys in their classes. None of these participants described that they did alternative activities together with the non-participating girls, at times when they could not participate in the scheduled physical education activities, and one even pronounced that that was not an option.

> Well, I'm a guy, so I'm still with the guys. I couldn't be the only guy being with the girls…that would be… that would stand out… It's perfectly fine, but it *would* stand out. (Participant 2, 17 years)

Besides girls, temporarily ill or injured pupils were frequently mentioned as not participating in physical education, but also those who were overweight or unfit. Pupils in the latter group were described as allies, someone to team up with when not able to participate.

**Assuming, responsibility and guilt.** Like many young persons, transitioning to adulthood, participants seemed to struggle to fit in. Several participants seemed eager to explain how they conformed to what seemed to be expected of them in terms of physical education and leisure activity, and one girl used the strategy to talk loudly about her work-out achievements, so that others could hear, and realize her physical potential. Further, several participants wanted to explain the reasons as to why they did not conform to expectations of exercise, such as reassuring their plans to start exercising shortly, but had been forced to prioritize schoolwork or rehabilitation among other things. Hence, voluntary or not, some resisted expectations to keep fit, but their resistance seemed to come at the cost of guilt or unease. Some participants limited themselves from participating in team sports according to what they described as being expected of them, but also by them, for the sake of the class or team. In this way they avoided stigmatization, meanwhile also conformed to prevailing expectations allowing particular bodies the reward of participation while excluding others. The following quote refers to a baseball-like game, commonly played as a social activity and in school-settings, which the participant wanted to join.

> …Anyway, I didn't participate…in this class-day, because I was thinking they won't have patience with me… and, it will become so intense… It was such a shame, really.
>
> Interviewer: …/…So you took on responsibility, for your class, that is…?
>
> Participant: [sighs] exactly…(Participant 3, 17 years)

Being loyal to their physical education teachers, who were in positions of power to provide inclusion, was another way that participants took on a personal responsibility. It was revealed through defending the teachers for giving lessons with lacking adaptation. Thereby responsibility for inclusive physical education was moved from the teacher and the school organization to the individual pupil.

> …well, they [physical education teachers] sort of couldn't do any better [to adapt class], but…it was kind of tough because it didn't become as social. (Participant 8, 17 years)

The responsibility the teenagers took upon themselves also indicated a sense that they had a limited right to complain about their situation. As one boy expressed it, comparing himself to others: *"I know that some people are worse off, sort*

*of"* (Participant 4, 17 years). Consequently, some described feeling that they ought to be content when they received support, regardless of its quality.

## Struggle to resist or conform to barriers—discussion category 2

In the second category, "Resisting prejudice and discrimination", participants struggled to resist or conform to barriers and norms, feeling somewhat more empowered than in the first category, yet still excluded in many ways. Despite insufficient support or adaptations, several participants defended teachers or others providing support, taking responsibility for inadequate participation upon themselves, as if not allowed to expect full inclusion. This responsibility often included shame, as noted by Jóhannsdóttir et al. [42], and sometimes led to self-exclusion, similar to findings by Haegele and Sutherland [18].

The current study, like Asbjørnslett et al. [43], shows that disabled youth want to be seen as 'ordinary' but recognize that inclusion requires hard work and that expectations differ for them compared to peers not seen as disabled. Not adhering to health discourses valuing certain bodies and behaviours can create guilt and shame, as experienced by youth in general [44]. When the body differs from both the ideal and what is considered 'normal', these ideals become even more unattainable, leading to internalized prejudice.

During childhood, several participants experienced not being listened to by adults, as also reported by youth with cerebral palsy in a study by Wintels et al. [45]. They were expected to need care due to being both children and disabled, limiting their perceived participation. Garland-Thomson [31] identified this infantilizing treatment as *the sentimental mode*, where individuals are seen as needing care. Meanwhile, Jahnsen et al. [7] found that learning personal responsibility for health in childhood strongly predicts regular physical activity in adulthood, which requires support and participation in decisions.

Participants' resistance to being labelled as disabled highlighted their struggle with norms and expectations. Their unwillingness to stand out reflects the stigma of having a different function and the effort to conform to bodily and behavioural expectations. Another way they minimized their otherness was by comparing themselves to those considered more disabled, in accordance with results reported by Wickman [46] and van Amsterdam et al. [47]. This positioning of others as deviant reproduced ableism, aligning with Garland-Thomson's [31] *exotic mode*.

Exclusionary practices regarding girls' non-participation in physical education were described, but none of the participants saw themselves as 'one of those girls'. While one participant noted boys' privilege in physical education due to their involvement in team sports during leisure time, most did not question girls' limited participation, or consider which pupils physical education was designed to include. Some even suggested that girls' non-participation was due to comfort or laziness. Thus, boys and masculinity dominated physical education and sports, while traits considered feminine were subordinate, which, as noted in previous studies [48,49], remained largely unquestioned.

## Category 3: Empowerment and support

The third category represents situations and environments in which participants described themselves as content with their physical education and leisure activity, even if sometimes preceded by a compromise, and where they were in power to influence their participation. All participants described such situations to different degrees and in different ways.

Several participants had found individual physical activities, such as horseback riding, biking, or running that worked out well for them. By adapting to norms and physical prerequisites, and by choosing activities less team-based and thereby less socially demanding, they did not need to struggle as much as was described in previous categories. Challenging normative expectations as to their limited physicality in these settings did not seem to invoke ambivalence or guilt but rather represented an active choice and a matured self-assurance.

> … just the fact that I dared to try new exercises [at the gym], and dared to pressure myself, I was like – shit, I'm strong! – I would never have done that five years ago. Participant 3, 17 years)

The above quote also described a sense of power that came with growing up, mentioned by several participants, that entailed confidence. Participants also voiced that growing up had given them an internal motivation propelled by a greater understanding of what positive effects physical activity could confer to them. The insights of benefits from physical activity also made participants appreciate the guidance and support they were provided, for example from physical education teachers or physiotherapists. The level of support received was described as an important part of being in charge of one's situation and experiencing empowerment. Hence, support was desirable when in accordance with their own wishes and needs, and particularly for being part of the group. During physical education, this support seemed particularly important, and one participant described what he regarded successful inclusion through active participation in physical education planning:

> So we sat there, planning together, for…eh, what would be possible to perform…/…and he [the physical education teacher] said like "here and here we'll be skiing and skating so I suggest this activity for you instead—how do you think that will work out?" "yes, I think that'll be fine"…Just something like that, really helps a lot. (Participant 7, 16 years)

It emerged from the interviews that being able to participate in social interaction with the class was important. One participant experienced inclusion when he got to oversee the secretariat during physical education, when the class was doing sports that he could not participate in.

> I couldn't play so I got to be the team coach in my class and such, I got to be the referee. I was not excluded but was just as included as the rest who were playing…/…that was nice. (Participant 1, 16 years)

In opposition to the experience of participant 1, another participant experienced the same type of assignment as degrading as he wanted to work with his body. Such discrepancies emphasized the importance of including the youth in planning the activities. Another positive example of inclusive activities was free-for-all ball games where different strategies could be used to succeed. One participant used kneepads, aimed to protect his knees if he fell, to avoid being hit by the ball.

> I remember, I used them to my advantage in Killer-ball in middle school, and it was so much fun to run, as I recall, so when someone threw the ball, I just threw myself to the floor, but I had the kneepads, so it didn't matter. (Participant 2, 17 years)

In a similar game, another participant described appreciating the fact that her peers were not being overly considerate toward her, as that made her feel included on equal terms. Moreover, recognizing oneself in others was mentioned as important for self-esteem, and participants who had at some time participated in physical education classes particularly designed for youth with physical impairments, described them as inclusive, yet adequately challenging.

**Support facilitates participation—discussion category 3.** In the last category, Empowered and with appropriate support, participants challenged norms and attitudinal barriers with confidence and decisiveness rather than guilt and shame. They chose to reject unwanted recommendations and embrace beneficial support, showing resistance through motivation to counter discouraging behaviours.

Being treated like others and receiving necessary support facilitated participation, in accordance with Brady et al. [50]. Teachers played a crucial role in fostering inclusion and reducing stigmatization. Shields and Synnot [21] describe successful inclusion as linked to teacher engagement, willingness to find solutions, and communication with joint decision-making, which was highlighted in several studies [21,51,52]. Ring et al. [39] further emphasized the need to negotiate physical education goals and outcomes collaboratively. Participants in the current study stressed how being listened to and receiving appropriate support facilitated participation in physical activities and education. Garland-Thomson [31]

described this as the *realistic mode*, emphasizing 'sameness' rather than impairments. The importance of support was also highlighted by Sandström et al. [23] and McLaughlin [53], who concluded that support should involve input from those receiving it, facilitated by recognizing support providers as valuable actors in society.

## Methodological considerations

The representation of physical leisure activity and physical education was unequally distributed within the three categories, likely due to their different characteristics. Physical education consists mainly of mandatory group activities, making it harder to opt out compared to physical leisure activities, and thus includes more situations where inclusion can fail. Our results may indicate that University-level physical education pedagogy needs further exploration to emphasize equity and inclusion, in accordance with Apelmo et al. [17]. Symeonidou et al. [54] found that continous seminars during teacher education, based on the social model of disability, entailed teachers to become more critical of oppressive approaches and implement anti-oppressive lessons.

The current study may have attracted physically active youth, potentially underrepresenting physically inactive youth with cerebral palsy. There was also an underrepresentation of youth with severe motor function difficulties and intellectual disabilities, limiting transferability to these groups. Our discussions with recruiting care providers about communication ability compatible with the interviewer's ability to understand, likely entailed that some youth with severe communication difficulties were excluded from invitation. These communication issues may also have created insecurity among recruiters that further restricted the selection of eligible youth invited to participate, which is unfortunate. Moreover, the interviewer's profession as a physiotherapist may have influenced participants to discuss physical activity in a way they thought was expected [55]. Participants did voice critiques, however, regarding physiotherapy as well as physical activity, suggesting limited impact.

Interview data was not returned to participants for checking, to maintain unedited material. As Sandelowski [56] suggests member checking may not ensure descriptive validity when interviews are recorded.

Participants were ethnically homogenous, and shared no reflections on barriers related to ethnicity or skin colour, despite interview questions about social injustices. This absence may reflect the unconscious privilege of conforming to the whiteness norm in Sweden [57]. Previous studies showed lower participation in physical education and leisure activity among children and youth with cerebral palsy born outside Sweden, particularly outside Europe [10]. Failure of the recruitment process to include these youth is concerning, and further studies targeting youth with minority roots are suggested to capture a broader picture of barriers to physical activity for youth with cerebral palsy.

## Conclusion

The aim of this study was to investigate the experiences of physical education and physical leisure activities among youth with cerebral palsy. It examines how these youths navigate contextual barriers and facilitators to participation, with a focus on norms of function and gender.

Our analysis revealed that many participants were treated as fundamentally different—as if the general rights and expectations afforded to youth did not apply to them—resulting in unequal opportunities in physical education and leisure activities. Yet, the examples they provided of effective communication and inclusive practices require awareness of limiting norms but still appear as relatively easy measures to take to prevent discrimination. Physical education was particularly problematic, with the teacher's approach being crucial to the participants' experiences. This type of discrimination suggests that schools may not be meeting their obligations to provide accessible education. Most barriers were linked to norms about function, regardless of its relevance to participation. The subordination of femininity in sports also posed an indirect barrier. Participants resisted limiting norms but also internalized negative preconceptions, experiencing shame. Interventions to promote social inclusion in physical education and organized leisure activities is recommended and requires awareness of norms and discrimination and a broader societal responsibility. All participants felt empowered

when treated equally and involved in decisions about support or adaptations needed for physical education and leisure activity participation.

## Acknowledgments

The authors want to acknowledge the youth who have kindly shared their thoughts and experiences, and the professionals in the child rehabilitation units throughout Sweden who helped us get in contact with these youth. We would also like to acknowledge the participants in our reference group that have helped us throughout the project.

## Author contributions

**Conceptualization:** Frida Degerstedt, Britt-Inger Keisu, Martin Björklund, Birgit Enberg.

**Data curation:** Frida Degerstedt.

**Formal analysis:** Frida Degerstedt, Britt-Inger Keisu.

**Funding acquisition:** Frida Degerstedt, Britt-Inger Keisu, Martin Björklund, Birgit Enberg.

**Methodology:** Frida Degerstedt, Britt-Inger Keisu, Birgit Enberg.

**Project administration:** Martin Björklund, Birgit Enberg.

**Resources:** Britt-Inger Keisu.

**Supervision:** Britt-Inger Keisu, Martin Björklund, Birgit Enberg.

**Validation:** Birgit Enberg.

**Writing – original draft:** Frida Degerstedt.

**Writing – review & editing:** Britt-Inger Keisu, Martin Björklund, Birgit Enberg.

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
