## [Decision Letter · Decision Letter 0]

1 Dec 2025

Dear Dr. Degerstedt,

Thank you for submitting your manuscript to PLOS ONE. After careful consideration, we feel that it has merit but does not fully meet PLOS ONE’s publication criteria as it currently stands. Therefore, we invite you to submit a revised version of the manuscript that addresses the points raised during the review process.

We look forward to receiving your revised manuscript.

Kind regards,

Efrem Kentiba, PhD

Academic Editor

PLOS ONE

Journal Requirements:

“FD, BIK: Umeå Center for Gender Studies at Umeå University; MB: the Linnea and Josef Carlsson Foundation; FD: The Queen Silvia’s Jubilee foundation 2019; and MB, BE: the Strategic Research Area Health Care Science (SFO-V), Umeå University.”

Please include this amended Role of Funder statement in your cover letter; we will change the online submission form on your behalf

Reviewers' comments:

Reviewer's Responses to Questions

**Comments to the Author**

1. Is the manuscript technically sound, and do the data support the conclusions?

Reviewer #1: Partly

Reviewer #2: Yes

2. Has the statistical analysis been performed appropriately and rigorously?

Reviewer #1: N/A

Reviewer #2: N/A

3. Have the authors made all data underlying the findings in their manuscript fully available?

Reviewer #1: No

Reviewer #2: No

4. Is the manuscript presented in an intelligible fashion and written in standard English?

Reviewer #1: Yes

Reviewer #2: Yes

Reviewer #1: Thank you for your efforts in trying to contribute to literature in this field. Kindly address the following comments in order to improve readability and clarity of your subject matter.

Title: A title should be concise and comprehensive to readership. It should always indicate the study design used. This is not the case with this title "Navigating Norms and Barriers: Youth with Cerebral Palsy in Physical Education and Leisure Activities in Sweden." Please, have a closer look at the title to make it more precise and to prompt reading, especially to people not in the field.

Introduction: What is the purpose and significance of this study? For the purpose of this study, could you explain the difference between physical education and physical leisure activities? Please indicate gaps from the literature review that this study intends to bridge.

- What are the relevant controversies in this field from your literature review?

Methodology: Could you justify the choice of methodology for this study?

- This sentence "Their physical function ranged from very limited ability to voluntary movement to very small motor limitations." is difficult to understand. For clarity purposes, I suggest you reformulate this sentence in a way that a lay person can grasp what you want to communicate.

- Lines 173-176 is not clear. Please clarify.

- Participants between 15 and 18 years in this study are minors and parental consent is an obligation. Was parental consent needed for any of the participants? The authors have not explicitly indicated anywhere in this section whether or not parental/guardian consent was duly obtained.

- Were the interviews done with the participants alone or in the presence of a parent/guardian? What measures were put in place to maintain comfort of the participants, considering their special conditions, during interviews? Was there any occasion that necessitated discontinuing the interview because a participant did not feel well or just stressed to continue? If yes, how was the situation handled?

- Patients with cerebral palsy are individuals with different skills and capabilities. What were the criteria for selecting participants for this study?

- What steps were taken to ensure reliability and triangulation of data?

- It is not enough to state "Ethical approval was obtained". What institutional body granted the approval?

Results and discussion:

- Some quotes from the participants and interviewer lack clarity and are too fragmented. For example, lines 392-396 are not comprehensible, especially to someone not familiar with the findings of the study. These quotes can't be reproduced because of the way they have been presented to the reader. Please go through the manuscript and ensure that the quotes make meaning, standing alone.

- Where are the quotes to show how or what participants said in lines 404-406? Also from lines 407-446 there is not a single quote from any participant. If this section is combined results and discussion, I expect that the discussion should be aligned to the findings from the study, stating the participants perceptions and feelings.

- Why is "Methodological considerations" part of the results and discussion section? Please explain.

Reviewer #2: The authors have taken great care to consider the ethics of talking with youth with cerebral palsy. There decision to not share their data publicly is sound to protect the confidentiality of participating youth, and they have offered the option of contacting the authors for access. This study is not one focused on replicability, in my view, it was more to understand the lived experiences of youth with cerebral palsy in participating in school sports. To that end, the need to look at the data would only be to understand the coding process/codes. The examples provided in the article do a good job of showing how the codes were created. It might be helpful to provide examples from each of the 3 categories in Table 1 (line 221 on page 10) since the data is not publicly available. Other than that minor suggestion, it was a well designed study with thoughful analysis.

.

Reviewer #1: **Yes:**Dr. Agnes Ebotabe ArreyDr. Agnes Ebotabe ArreyDr. Agnes Ebotabe ArreyDr. Agnes Ebotabe Arrey

Reviewer #2: **Yes:**Neeraja AravamudanNeeraja AravamudanNeeraja AravamudanNeeraja Aravamudan

---

## [Author Response · Author response to Decision Letter 1]

6 Jan 2026

We are grateful for your valuable comments. They have helped improve this manuscript. Please, find our detailed responses to each comment in the attached file "Response to reviewers".

---

## [Decision Letter · Decision Letter 1]

31 Jan 2026

Dear Dr. Degerstedt,

Thank you for submitting your manuscript to PLOS ONE. After careful consideration, we feel that it has merit but does not fully meet PLOS ONE’s publication criteria as it currently stands. Therefore, we invite you to submit a revised version of the manuscript that addresses the points raised during the review process.

We look forward to receiving your revised manuscript.

Kind regards,

Efrem Kentiba, PhD

Academic Editor

PLOS One

Journal Requirements:

Reviewers' comments:

Reviewer's Responses to Questions

**Comments to the Author**

Reviewer #1: (No Response)

Reviewer #2: All comments have been addressed

2. Is the manuscript technically sound, and do the data support the conclusions?

Reviewer #1: Partly

Reviewer #2: Yes

3. Has the statistical analysis been performed appropriately and rigorously?

Reviewer #1: N/A

Reviewer #2: N/A

4. Have the authors made all data underlying the findings in their manuscript fully available?

Reviewer #1: No

Reviewer #2: No

5. Is the manuscript presented in an intelligible fashion and written in standard English?

Reviewer #1: Yes

Reviewer #2: Yes

Reviewer #1: Thank you for your efforts in trying to address the concerns and comments raised. However, comments no 6 and 18 still need to be addressed.

The title (comment 6) is more descriptive and understandable but should indicate where the study was done, in this case Sweden.

Your response to comment 18 "We chose not to add a quote to lines 404-406 (final paragraph of category 2) as there was no data suitable to form a comprehensible quote that would provide clarity to the analysis. Please note that this does not equal non-empirical analysis" does not address the comment. It is not what you choose to add or not. Your study data should provide evidence to draw conclusions from. If "there was no suitable data to form a comprehensive quote", then why make any statement that can't be substantiated from the interviews that you conducted? Please revisit your interviews to make an informed decision on what to add or not.

Reviewer #2: You have addressed all of the reviewer feedback effectively, strengthening the paper's methodology and discussion.

.

Reviewer #1: **Yes:**Dr Agnes Ebotabe ArreyDr Agnes Ebotabe ArreyDr Agnes Ebotabe ArreyDr Agnes Ebotabe Arrey

Reviewer #2: No

---

## [Author Response · Author response to Decision Letter 2]

6 Feb 2026

Thank you for reviewing this manuscript, now named Barriers and facilitators in physical activity among youth with cerebral palsy: A qualitative study in Sweden. Your useful comments are addressed in detail below. We believe they have improved the manuscript greatly. Page- and line numbers refer to when the track-changes are invisible. The responses are also added as a separate file.

Reviewer #1: Thank you for your efforts in trying to address the concerns and comments raised. However, comments no 6 and 18 still need to be addressed.

The title (comment 6) is more descriptive and understandable but should indicate where the study was done, in this case Sweden. Thank you, this is now corrected. The following title is now used Barriers and facilitators in physical activity among youth with cerebral palsy: A qualitative study in Sweden

Your response to comment 18 "We chose not to add a quote to lines 404-406 (final paragraph of category 2) as there was no data suitable to form a comprehensible quote that would provide clarity to the analysis. Please note that this does not equal non-empirical analysis" does not address the comment. It is not what you choose to add or not. Your study data should provide evidence to draw conclusions from. If "there was no suitable data to form a comprehensive quote", then why make any statement that can't be substantiated from the interviews that you conducted? Please revisit your interviews to make an informed decision on what to add or not. Thank you for your thorough comment. The text section is revised (p. 20, lines 441-5), and a brief quote is added to exemplify the analysis. I hope this contributes with clarity. The responsibility the teenagers took upon themselves also indicated a sense that they had a limited right to complain about their situation. As one boy expressed it, comparing himself to others: “I know that some people are worse off, sort of” (Participant 4, 17 years). Consequently, some described feeling that they ought to be content when they received support, regardless of its quality.

Reviewer #2: You have addressed all of the reviewer feedback effectively, strengthening the paper's methodology and discussion. Thank you for your time and your previous comments and advice

---

## [Decision Letter · Decision Letter 2]

11 Mar 2026

Dear Dr. Degerstedt,

Thank you for submitting your manuscript to PLOS ONE. After careful consideration, we feel that it has merit but does not fully meet PLOS ONE’s publication criteria as it currently stands. Therefore, we invite you to submit a revised version of the manuscript that addresses the points raised during the review process.

We look forward to receiving your revised manuscript.

Kind regards,

Efrem Kentiba, PhD

Academic Editor

PLOS One

Journal Requirements:

Reviewers' comments:

Reviewer's Responses to Questions

**Comments to the Author**

Reviewer #1: (No Response)

2. Is the manuscript technically sound, and do the data support the conclusions?

Reviewer #1: Yes

3. Has the statistical analysis been performed appropriately and rigorously?

Reviewer #1: Yes

4. Have the authors made all data underlying the findings in their manuscript fully available?

Reviewer #1: Yes

5. Is the manuscript presented in an intelligible fashion and written in standard English?

Reviewer #1: Yes

Reviewer #1: Thank you for taking time to address my comments. However, the title still needs to be rephrased.

should read "Barriers... cerebral palsy in Sweden: a qualitative study". The study is about participants living in Sweden and not qualitative study in Sweden. Refer to your previous study: Degerstedt F, Björklund M, Keisu B-I, Enberg B. Unequal physical activity among children with cerebral palsy in Sweden—A national registry study.

.

Reviewer #1: **Yes:**Dr Agnes Ebotabe ArreyDr Agnes Ebotabe ArreyDr Agnes Ebotabe ArreyDr Agnes Ebotabe Arrey

---

## [Author Response · Author response to Decision Letter 3]

13 Mar 2026

Thank you for your thorough revision. For clarity I below include the previous correspondence referring to the title. It is now corrected according to your latest suggestion in bullet point 4 below, and it also aligns with journal guidelines. Thank you for helping refine the precision of the title, I hope you find it improved in clarity.

1. Original title: Navigating Norms and Barriers: Youth with Cerebral Palsy in Physical Education and Leisure Activities in Sweden

Comment: Title: A title should be concise and comprehensive to readership. It should always indicate the study design used. This is not the case with this title "Navigating Norms and Barriers: Youth with Cerebral Palsy in Physical Education and Leisure Activities in Sweden." Please, have a closer look at the title to make it more precise and to prompt reading, especially to people not in the field.

2. Revision 1: Barriers and facilitators in physical activity among youth with cerebral palsy: A qualitative study

Comment: The title (comment 6) is more descriptive and understandable but should indicate where the study was done, in this case Sweden.

3. Revision 2: Barriers and facilitators in physical activity among youth with cerebral palsy: A qualitative study in Sweden

Comment: Thank you for taking time to address my comments. However, the title still needs to be rephrased. should read "Barriers... cerebral palsy in Sweden: a qualitative study". The study is about participants living in Sweden and not qualitative study in Sweden. Refer to your previous study: Degerstedt F, Björklund M, Keisu B-I, Enberg B. Unequal physical activity among children with cerebral palsy in Sweden—A national registry study.

4. Revision 3: Barriers and facilitators in physical activity among youth with cerebral palsy in Sweden: a qualitative study

---

## [Editor Report · Decision Letter 3]

22 Mar 2026

Barriers and facilitators in physical activity among youth with cerebral palsy in Sweden: A qualitative study

PONE-D-25-48951R3

Dear Dr. Degerstedt,

We’re pleased to inform you that your manuscript has been judged scientifically suitable for publication and will be formally accepted for publication once it meets all outstanding technical requirements.

Within one week, you’ll receive an e-mail detailing the required amendments. When these have been addressed, you’ll receive a formal acceptance letter, and your manuscript will be scheduled for publication.

Kind regards,

Efrem Kentiba, PhD

Academic Editor

PLOS One
---

## [Editor Report · Acceptance letter]

PONE-D-25-48951R3

PLOS One

Dear Dr. Degerstedt,

I'm pleased to inform you that your manuscript has been deemed suitable for publication in PLOS One. Congratulations! Your manuscript is now being handed over to our production team.

Kind regards,

on behalf of

Dr. Efrem Kentiba

Academic Editor

PLOS One